# Virological and Genetic Characterization of the Unusual Avian Influenza H14Nx Viruses in the Northern Asia

**DOI:** 10.3390/v15030734

**Published:** 2023-03-11

**Authors:** Nikita Dubovitskiy, Anastasiya Derko, Ivan Sobolev, Elena Prokopyeva, Tatyana Murashkina, Maria Solomatina, Olga Kurskaya, Andrey Komissarov, Artem Fadeev, Daria Danilenko, Polina Petrova, Junki Mine, Ryota Tsunekuni, Yuko Uchida, Takehiko Saito, Alexander Shestopalov, Kirill Sharshov

**Affiliations:** 1Federal Research Center of Fundamental and Translational Medicine, 630060 Novosibirsk, Russia; 2Smorodintsev Research Institute of Influenza, 197022 Saint Petersburg, Russia; 3National Institute of Animal Health, Tsukuba 305-0856, Japan

**Keywords:** Influenza A virus, Avian influenza, H14 subtype, H14N9, Siberia, wild birds, waterfowl, Northern Asia

## Abstract

Wild aquatic birds are generally identified as a natural reservoir of avian influenza viruses (AIVs), where a high diversity of subtypes has been detected. Some AIV subtypes are considered to have relatively low prevalence in wild bird populations. Six-year AIV surveillance in Siberia revealed sporadic cases of the rarely identified H14-subtype AIV circulation. Complete genome sequencing of three H14 isolates were performed, and the analysis indicated interconnections between low pathogenic avian influenza (LPAI) viruses. We conducted hemagglutination inhibition and virus neutralization assays, estimated the susceptibility of isolates to neuraminidase inhibitors, and characterized receptor specificity. Our study revealed circulation of a new H14N9 subtype described for the first time. However, the low prevalence of the H14-subtype AIV population may be the reason for the underestimation of the diversity of H14-subtype AIVs. According to the available data, a region in which H14-subtype viruses were detected several times in 2007–2022 in the Eastern Hemisphere is Western Siberia, while the virus was also detected once in South Asia (Pakistan). Phylogenetic analysis of HA segment sequences revealed the circulation of two clades of H14-subtype viruses originated from initial 1980s Eurasian clade; the first was detected in Northern America and the second in Eurasia.

## 1. Introduction

Avian influenza viruses belong to *Alphainfluenzavirus influenzae* species of the Orthomyxoviridae family based on the proposed ICTV taxonomy. Considerable diversity of *Alphainfluenzavirus influenzae* virus subtypes with respect to hemagglutinin (HA) (H1-H16) and neuraminidase (NA) (N1-N9) proteins has been detected in wild waterfowl, while the subtypes H17, H18, N10 and N11 were exclusively detected in bats [1,2,3]. Highly pathogenic avian influenza (HPAI) viruses have an impact on both the poultry industry and the wild bird population, and can be dangerous for human health [4].

Wild waterfowl are considered to be a natural reservoir of avian influenza viruses, and can carry viruses over long distances during seasonal migration [5]. Up until October 2022, more than 52,000 genome sequences of avian influenza strains had been submitted to the EpiFlu GISAID database (accessed 26 October 2022). Nevertheless, the H14 sequences comprised only 0.089% (n = 46, duplicated submissions excluded, strains from research included) of that amount, collected over 40 years of overall observation. Historically, three isolates of the H14 subtype were first described in the Soviet Union in 1982, collected from mallards and herring gull (A/Mallard/Gurjev/244/82 (H14N6), A/mallard/Gurjev/263/1982, and A/herring gull/Astrakhan/267/1982) [6]. Since then, only three strains of the H14 subtype have been detected in Eurasia: H14N6, isolated from garganey (*Anas querquedula*) in Ukraine in 2006, H14N3, isolated from goose in Pakistan in 2014, and H14N7, isolated from sandpiper in Tomsk by Marchenko et al. in 2019 [7]. There has only been one report of the AIV H14 subtype being detected in Eurasia since 2015, whereas in North America, 36 isolates were obtained between 2010 and 2018 [7,8,9,10]. Hemagglutinin-neuraminidase (HA-NA) constellations of known isolates are limited to six NA subtypes: H14N3, H14N4, H14N5, H14N6, H14N7, and H14N8.

Avian influenza virus (AIV) surveillance in Russia revealed three new H14-subtype variants of avian influenza viruses circulating in Eurasia, whereas only one complete genome sequences of H14 AIV strains from Russia from the last 39 years is stored in the GISAID database. Here, we describe a new genetic subtype of AIV—H14N9. The aim of the present study is to genetically characterize the rarely identified avian influenza H14Nx viruses in Siberia, including the H14N9 strain, which has been isolated for the first time, and to analyze the virological characteristics of these isolates.

## 2. Materials and Methods

### 2.1. Sampling

Samples from wild birds were collected according to international ethical standards and the national legislation of the Russian Federation. Samples from duck species were collected during hunting seasons with a license from the regional Ministries of Ecology and Natural resources under the program for surveillance of infectious diseases in wild animal populations (FRC FTM, Novosibirsk).

Cloacal swabs were collected from wild birds in the Asian part of Russia between 2014 and 2019. Sampling sites are presented on the map, constructed using the R packages rnaturalearth, sf, and ggplot2 (Figure 1). Samples were stored in liquid nitrogen in individual tubes filled with sterile transport medium before delivery to the laboratory.

### 2.2. Virus Detection, Isolation, and RNA Extraction

The cloacal swab medium supernatant of each sample was inoculated (200 μL) into three 10-day specific-pathogen-free (SPF) embryonated chicken eggs using standard methods described elsewhere [11,12]. Virus titers in the harvested allantoic fluid were measured using hemagglutination assay (HA) as described in the standard protocol [13].

Extraction of viral RNAs from allantoic fluid samples was conducted using the RIBOsorb RNA extraction kit (AmpliSens, Moscow, Russia) following reverse transcription using the Reverta-L kit (AmpliSens, Moscow, Russia). PCR for the detection of influenza virus nucleic acid in isolated samples was performed using AmpliSens Influenza virus A/B-FL kit (AmpliSens, Moscow, Russia) according to the manufacturer’s instructions.

### 2.3. Virological Characteristics

To provide virological characterization of strains in the study, two representative strains of phylogenetically different subclades were chosen: A/Common Teal/Chany_Lake/29/2019 (H14N3) (A/29) and A/garganey/Chany Lake/210/2014 (H14N9) (A/210). An additional strain, A/shoveler/Omsk/15O/2019 (H4N6), was added to the experiment, as the representative strain of the sister H4 subtype according to Kawaoka et al., 1990 [6]. We also used A (H1N1)pdm09 as a reference group for oseltamivir sensitivity tests.

#### 2.3.1. Cell Cultures

The 50% tissue culture infectious dose (TCID_50_) for MDCK cells was determined for A/29, A/210 and A/15O. For this purpose, serial 10-fold dilutions of viruses were performed, and 96-well plates with confluent MDCK cell monolayer were inoculated with dilutions of virus. All virus titrations were performed in duplicate. After 30 min of incubation, the supernatant was removed, and MEM with 0.2% BSA and 2 µg/mL of trypsin was added to one plate and MEM with 2% FBS was added to another plate with the same virus. Plates were incubated at 37 °C and 5% CO_2_ for 5 days. Virus-induced cytopathic effect was detected, and virus titers were determined as the 50% tissue culture infectious dose (TCID_50_) per ml according to the Kerber method with Ashmarin–Vorobyov modification [14] as follows: log_10_TCID_50_/mL = lgDn − δ(ΣLi − 0.5); where:

Dn—maximum effect of dose;

Li—the ratio of the number of wells with cytopathyic effect to the total number of wells infected with this dose;

i—number of dose;

δ—the logarithm of virus dilutions.

#### 2.3.2. Hemagglutination Inhibition and Neutralization Assay

To obtain polyclonal sera for antigenic analysis, 6-week-old specific-pathogen-free quails were intravenously boosted with 0.5 mL with 10^6.0^ EID_50_ of the virus. After 7 days, the quails were re-inoculated. On day 14 post inoculation (p.i.), blood samples were collected, and sera were harvested for antigenic analysis.

Antigenic analysis of A/H14 strains was performed in hemagglutination inhibition (HI) and virus neutralization (VN) tests using quail polyclonal antisera obtained as described above. Before testing, all sera samples were heated at 56 °C for 30 min. Tests were performed according to standard protocols. The highest dilution of the serum that completely inhibited hemagglutination or neutralized virus growth was taken as the HI/VN titer [15]. Viruses were considered antigenically similar if their HI/VN titer differences represented no more than a two-fold dilution.

#### 2.3.3. Determination of Susceptibility to Neuraminidase Inhibitors

The susceptibility of the H14N3, H14N9, and H4N6 strains to oseltamivir carboxylate (Hoffmann-La Roche, Basel, Switzerland) was evaluated by published NA inhibition assays [16,17]. Briefly, viruses were standardized to an NA activity level 10-fold higher than that of the background, as measured by the production of fluorescent product from methylumbelliferyl-N-acetylneuraminic acid (MUNANA) substrate (Sigma-Aldrich, Darmstadt, Germany). Drug susceptibility profiles were determined by the extent of NA inhibition (NAI) after incubation with 3-fold serial dilutions of NAIs. The 50% inhibitory concentrations (IC50) were determined from the dose–response curve.

#### 2.3.4. Receptor Specificity

The receptor specificity of the virus was characterized by determining the level of binding to glycoconjugates immobilized on a microarray (Semiotic, Moscow). Briefly, the microarrays were blocked with a standard blocking buffer (50 mM Ethanolamine (pH 8.5): 400 mL dH_2_O, 1195 µL ethanolamine, 3.8 g boric acid, 800 µL Tween 20) for 2 h with constant stirring at room temperature. After that, the arrays were washed in PBS-Tween 20 0.05% twice. The titer of the test virus was determined in HA assay in the presence of a neuraminidase inhibitor (oseltamivir carboxylate); the virus was then diluted to obtain 16 HA units (buffer for dilution: PBS supplemented with 0.01% Tween20 and 20 μM oseltamivir carboxylate). Next, 1 mL of 16 HAU of the virus was applied on the microarray. The arrays were incubated for 16–18 h at 4 °C. After incubation, the arrays were washed twice with buffer (PBS + 0.05% Tween 20). Then, the arrays were covered with 10 M biotinylated polyacrylamide conjugate with 3′SLN (Neu5Acα2–3Galβ1–4GlcNAcβ) in PBS with an inhibitor followed by incubation for 1 h at 4 °C and washing of the arrays with buffer (PBS + 0.05% Tween 20). Subsequently, the solution of fluorescently labeled streptavidin–Alexa 555 at a concentration of 2 μg/mL in PBS + 0.05% Tween 20 was applied. The arrays were incubated for 45 min at room temperature, washed twice with buffer (PBS + 0.05% Tween 20), then washed with distilled water, air dried, and scanned on a reader (Innoscan 710, Innopsys, Carbonne, France). The scanning results were processed in the ProScan Array Express program, with fluorescence intensity values expressed in relative fluorescence units (RFU) in the resulting tables. Images were scanned at a resolution of 10 µm. The selection of scanning parameters was carried out empirically for glycoarrays, taking into account the main requirement-more than 95% of the signals should fall within the measurement scale of the device, i.e., not to exceed an RFU value of 65,535. Then, the obtained images were imported into an Excel spreadsheet using the reader software and the .GAL file.

### 2.4. Sequencing and Genetic Characterization

Complete genome NGS sequencing of A/210 and A/211 strains was performed using the Illumina MiSeq platform and the associated reagent kits, also from Illumina, according to the methodology described by the manufacturer. RNA was extracted using QIAamp Viral RNA Mini Kit (QIAGEN, Hilden Germany). Whole-genome amplification was performed according to the protocol described by Zhou et al. [18]. DNA libraries were prepared using a Nextera DNA Flex Library Prep kit (Illumina, San Diego, CA, USA). Sequencing of the DNA libraries was conducted with a reagent kit, version 3 (600-cycle), on a MiSeq genome sequencer (Illumina). Read mapping was performed using minimap2 v2.17 software with default settings. Consensus sequences were obtained using SAMtools-mpileup v1.10 software.

A/29 strain library preparation was performed in the Animal Influenza unit of National Institute of Animal Health (Tsukuba, Japan), as described previously [19]. Briefly, isolation of RNA from allantoic fluid was performed using the RNeasy Mini kit (QIAGEN, Hilden, Germany). We used the NEBNext Ultra II RNA Library Prep Kit for Illumina (New England Biolabs, Ipswich, MA, USA) to prepare the cDNA libraries. Before sequencing, 10 pM of libraries were mixed with 10 pM of PhiX Control V3 (Illumina). Sequencing was performed using MiSeq genome sequencer (Illumina) using the MiSeq Reagent Kit v.2 (Illumina). Consensus sequences were constructed using Workbench software v.9.5.3 (QIAGEN, Germany).

We utilized the MUSCLE algorithm to acquire multiple alignments of sequences for H14 gene segments from available datasets, including EpiFlu (accession-June, 2022). Phylogenetic trees were evaluated using the IQTREE software and the maximum-likelihood algorithm with the best-fitting substitution model (ModelFinder option) and a bootstrap of 1000 iterations [20]. Molecular dating was performed using the TempEst 1.5.3 and BEAST v2.7.0 applications under an optimized relaxed molecular clock, coalescent Bayesian skyline tree prior, and a GTR + G4 + I model [21]. Two independent MCMC chains were run for 100 million generations each, and the first 10% were discarded as burn-in. Tracer was used to check convergence of the chains. Results were summarized using LogCombiner v 2.7.0. Tree visualization was performed using FigTree v1.4.4 software and MEGA11 software [22]. Average pairwise distances of amino acid sequences were estimated using MEGA11 software.

## 3. Results

### 3.1. Virus Isolation

In total, 5777 samples were collected from wild birds in the Asian part of Russia during the Avian Influenza surveillance program between 2014 and 2019 (Figure 1).

The overall isolation rate of all AIVs in this period amounted to 3.34% (n = 193), with only 0.05% (n = 3) corresponding to the detection of H14 cases. At the same time, the isolation rate of the most prevalent subtypes, H3Nx and H4Nx, remained constantly high throughout the time course of observation, comprising 1.45% (n = 84) and 0.38% (n = 22), respectively. Thus, the proportion of H14 viruses was significantly lower than viruses of the commonly presented subtypes. Specifically, during surveillance in wild waterfowl in 2014, 303 samples were examined that demonstrated the presence of AIVs, corresponding to 4.29%, whereas the H14 subtype was detected in 0.66% of cases (n = 2). The presence of AIVs among the 2366 samples collected in 2019 was 3.08% (n = 73), while the H14 subtype accounted for 0.04% (n = 1) (Table 1). The information provided here highlights the fact that H14-subtype viruses are uncommon in the studied territories of Northern Asia. In addition, the presence of this subtype in the “Siberian influenza virus pool” is low. The contribution of Siberian H14-subtype virus strains to the Eurasian H14 pool was the most dramatic (4/9). However, the distribution of overall detections of H14 in Eurasia in time is irregular (n = 3 in 1982, n = 1 in 2006, n = 3 in 2014, and n = 2 in 2019), suggesting sporadic transmission to migratory bird species involved in the surveillance program. One isolate of the H14 subtype from common teal (*Anas crecca*) and two isolates from garganey (*Anas querquedula*) were obtained in the autumn season at Chany Lake in the Novosibirsk region (Western Siberia, Russia): A/common teal/Chany_Lake/29/2019 (H14N3), A/garganey/Chany Lake/210/2014 (H14N9), and A/garganey/Chany Lake/211/2014 (H14N9), hereafter referred to as A/29, A/210, and A/211, respectively (Appendix A). Previously, the undescribed novel subtype H14N9 was isolated twice in 2014 from garganey. Additionally, we found and included in the study another novel H14 virus isolation case from sandpiper (*Calidris* sp.) in the neighboring Tomsk region previously described by Marchenko et al. [7]. Its sequence is currently available in the GISAID database (GISAID # 390458).

### 3.2. Virological Characteristics

#### 3.2.1. Growth in Cell Culture, Antigenic Analysis and Neuraminidase Inhibition Tests

For all tested viruses, the 50% tissue culture infectious dose was determined with and without the presence of trypsin. All analyzed viruses efficiently replicated in MDCK cells in similar titers (Table 2). All viruses also replicated efficiently in the absence of trypsin, although they had titers that were lower by an order of magnitude than those replicated in the presence of trypsin.

We evaluated the neuraminidase activity in vitro. In our experiment, all viruses (H14N3, H14N9, and H4N6) were highly sensitive to oseltamivir (Tamiflu) (Table 2). Oseltamivir is an inhibitor of the neuraminidase enzyme activities of influenza A and B viruses.

To determine the antigenic differences between A/H14 viruses, we performed antigenic analysis using polyclonal quail antisera raised against these viruses (Table 3). All H14 isolates demonstrated cross-reactivity with all quail polyclonal antisera and did not demonstrate cross-reaction with H4-antiserum.

#### 3.2.2. Receptor Specificity

In order to characterize the receptor binding profile of the isolated strains, commercially available glycoarrays were used. For comparison, the strain A/little tern/Guriev/779/83 (H16N3) was used, the receptor binding profile of which has been described previously. The analysis of the receptor specificity of the virus showed that the most preferred receptors for binding with the A/garganey/Chany Lake/210/2014 (H14N9) virus were sulfated structures. In all cases of interaction of the virus with glycans, the presence of a sulfate group was observed. The H14N9-subtype virus was not shown to interact with human-type receptors containing Neu5Acα2-6Gal, in contrast to the A/little tern/Guriev/779/83 (H16N3) virus, which showed binding to various receptor variants containing a link at positions 2–6, which is consistent with the data presented in the literature [23,24]. According to the results of this study, the A/little tern/Guriev/779/83 (H16N3) virus showed a high level of binding to the 3-SiaTn receptor, while the H14-subtype viruses did not bind these types of glycans at all (Table 4).

Interestingly, the H14N9 virus practically did not bind to the classical avian-type receptors-3′SL and 3′SLN.

### 3.3. Genetic Characterization and Phylogenetic Analysis

Illumina sequencing resulted in 146,880 paired-end reads for A/210 and 117,551 for A/211. A total of 138,552 and 108,379 paired-end reads passed the quality control pipeline for A/210 and A/211, respectively. Assemblies of complete genome sequences of eight gene segments for three isolates (A/210, A/211, A/29) were obtained: PB1 (2340, 2340, 2323 nt), PB2 (2340, 2340, 2323 nt), PA (2232, 2232, 2219 nt), HA (1747, 1747, 1721 nt), NP(1564, 1564, 1538 nt), NA (1457, 1457, 1433 nt), MP (1026, 1026, 1005 nt), NS (889, 889, 855 nt).

Identity analysis (BLAST) of HA segment nucleotide sequences showed that the nearest sequence for isolates A/210 and A/211 was that obtained from a virus isolate of the H14N3 subtype collected in 2014 in Pakistan. Isolate A/29 had the highest identity percentage with a strain isolated from sandpiper in 2014 in Siberia, which was the most recent sequence collected in Eurasia (Table 5). The first sequence of HA belonging to the H14 subtype, described in 1982 in the former USSR, shared 88.06, 88.88, and 89.00% identity with the studied A/29, A/210, A/211 sequences obtained in Russia, respectively. The identity analysis of each segment of the characterized isolates is provided in the Appendix A.

The results provided by Fries et al. (2013) for HA segment sequences of H14 subtype for the periods before 1989 and after 2009 showed overall mean pairwise distance at the level of 0.048 (SE = 0.009). The overall mean pairwise distance for the H14-subtype HA sequences, including the sequences submitted after 2009, is 0.029 (SE = 0.004), which is close to the estimation of phylogenetically related and frequently detected H4-subtype sequences (≤1982 and ≥2022) (0.030, SE = 0.004). These data support the assumptions of Fries et al. (2013) that H14-subtype viruses can circulate in spatial or temporal areas that are not involved in avian influenza surveillance [8].

All known H14-subtype AIVs can be phylogenetically divided into one of two clades on the basis of their HA segment nucleotide sequences: H14.1 and H14.2 (Figure 2). Clade H14.1 includes only those strains originally discovered in 1982 in the former USSR. The H14.2 clade includes the rest of the known strains isolated between 2006 and 2019. The latter consists of two subclades: H14.2.1 (Eurasian strains 2014–2019) and H14.2.2 (North American strains and a related strain from Ukraine, 2006). With respect to time-scaled phylogeny, the H14.2.1 subclade diverged from its most recent common ancestor with the H14.2.2 subclade in 2003 (95% HPD 2002–2005) (Appendix A). In this study, isolates from Russia were determined to belong to the H14.2.1 subclade of the H14 subtype, forming two subclades: H14.2.1a (2014–2015) and H14.2.1b (2019). Strains A/210 and A/211 of the H14N9 subtype belong to subclade H14.2.1a, along with the closely related H14N3 strain from Pakistan 2014. Strain A/29 of the H14N3 subtype belongs to the H14.2.1b subclade, and forms a monophyletic group with the previously described H14N7 strain from Tomsk, Russia 2019.

The nucleotide sequence of the NA segment of the N3 subtype was related to the H7N3-, H11N3-, and H10N3-subtypes of LPAI viruses isolated in 2018–2020 in distant territories of Eurasia (Figure 3).

The phylogenetic tree for the N9 segment sequences shows association with H7N9-H11N9-subtype strains from Europe, South and East Asia, and Northern Africa (Figure 4).

The phylogenetic tree for the NS segment points to the relatedness between the H14N3, H14N9, and two LPAI virus subclades of allele A from China, Mongolia, Northern Africa, and Bangladesh (Appendix A). The H14N3 strain is closely related to the H14N3 strain found in Pakistan in 2014. Another Russian H14 strain from Tomsk is grouped with the European strains, which include H5N1. Notably, among all of the H14 strains characterized, only seven variants belong to allele B of the NS segment. Of these seven strains, only one H14 strain can be attributed to the Eurasian clade, which was isolated in Ukraine in 2006, whereas the remaining six strains belong to the North American viruses of allele B. The HA segment of this variant is thought to possess the most recent common ancestor with the strains introduced to North America.

The PB2 of two H14N9 isolates (A/210, A/211) is clustered with two different clades, whereas all other segments belong to the same clusters, and do not show diversity. The PB2 of the A/211 strain is closely related to the H14N3 strain from Pakistan. Remarkably, PB2 from the H14 Tomsk strain is grouped together with the H5N1 strains that have been circulating in Europe since 2020 (Figure 5A).

The PB1 segment of strain A/29 has a common ancestor with H5N8-subtype variants that caused outbreaks in Europe during spring 2020 (Figure 5B) [25,26]. The PA segment sequences are attributed to LPAI viruses from China, Mongolia, Bangladesh, and Russia (Figure 5C). The PA segment of the Pakistan strain is clustered with the China clade, which is different from the Russian strain.

The NP segment of the H14N3 isolate is monophyletic with isolates from Bangladesh and Mongolia, similar to MP, while the H14N9 NP sequences are clustered with H1N1, H6N8, H10N5, and H5N5, which were acquired from wild waterfowl in Eastern Russia and Europe (Figure 5D). The MP sequence of H14N3 is closely related to the Eurasian lineage, particularly to strains of the H5N3, H11N9, H3N1, H6N7, and H4N6 subtypes from Mongolia, China, and Bangladesh that form the Central-Southwest Asian subclade (Figure 5E). Both H14N9 and H14N7 (Tomsk) MP segments cluster into the diverse subpopulation grouping with North African and European isolates from the period between 2015 and 2021. Geography of the spread of related strains from the Eastern to the Western part of the Eurasian continent (Figure 5D,E).

All of the information described above suggests that the evolution of H14-subtype viruses, accompanied by antigenic drift and plural reassortment events, occurs primarily in connection with reservoirs of LPAI viruses of different subtypes.

Analysis of the HA gene sequences reveals five and six amino acid substitutions for A/210 and A/211, respectively (Table 6). The A/211 isolate carries D326G substitution in the HA cleavage site, which is the only difference between it and the A/210 isolate HA amino acid sequence, and may be as a result of cultivation. The most recent H14 strain A/29 has nine substitutions compared to the A/goose/Karachi/NARC-13N-969/2014 (H14N3) strain, with A329T substitution in the HA cleavage site. Amino acid signature residues Q226 and G228 in the receptor-binding domain (RBD) of HA for the A/29, A/210, A/211 strains indicate avian receptor susceptibility [18].

A comprehensive analysis was performed of NAI resistance-associated substitutions of neuraminidase protein among the studied AIVs with N3 and N9 neuraminidase subtypes. We did not find any known NAI resistance-associated substitutions in A/29, A/210 and A/211 viruses (E119, Q136, G147, I222, L223, A246, H274, R292, R371, I427 (N2 numbering)) [27]. These results were confirmed for A/29 and A/210 in vitro by the low resistance of viruses to oseltamivir.

## 4. Discussion

Wild waterfowl are a natural reservoir of type A influenza viruses. Cases of transmission of avian influenza viruses to other hosts—pigs, humans, horses, etc.—have periodically been recorded, with reports having intensified in recent years [28], raising awareness of the origin of future influenza pandemic. The reasons for which some AIV subtypes are more common in some bird species than others remain insufficiently clear. Here, we describe the rarely identified H14Nx avian influenza viruses found in Siberia. The sequencing of samples collected from wild birds demonstrated the presence of a novel combination of the HA-NA constellation in the circulating LPAIs-H14N9 subtype. The low frequency of detection of the H14 subtype is a limiting factor in the investigation of the interrelations between virus subtype, bird species, and their geographical co-occurrence. However, the internal segments of the H14 subtype are thought not to be associated with subtype, bird species, or location. Phylogenetic reconstruction suggests that the evolution of internal segments most likely includes reassortment and antigenic drift mechanisms that provide a wide range of host adaptation. Remarkably, we and other authors have isolated H14-subtype viruses only from *Anseriformes* and *Charadriiformes* species, presenting at low rates. One previously published work hypothesizes that sampling biases in surveillance can form a blind spot for particular subtypes, for example, as a result of the remoteness of some territories [8]. Therefore, rare H14 detection may be the result of spillovers in cases where the virus invades surveyed populations from non-surveyed host reservoirs. The abundant isolation of H14 virus variants during certain periods in North America since 2010 could be an argument in favor of this hypothesis. Broadening the inventory by adding the list of non-surveyed species could help in testing this hypothesis. The second hypothesis (Fries et al., 2013) assumes cross-reactive immunity among subtypes H4–H14, resulting in possible antigenic evolutionary pressure of H4 LPAIs directed to H14-subtype avian influenza viruses. Subtype H4 is known to be the sister subtype of H14, having diverged from the most recent common ancestor [8]. To test this hypothesis, large-scale serological assays should be applied in future. In our study, we used polyclonal sera obtained for antigenic analysis and showed that all of the studied H14 isolates demonstrated antigenic similarity with one another on the basis of hemagglutination inhibition and neutralization tests, but did not exhibit significant cross-reaction with the H4 reference strain, thus confirming the difference from the sister subtype. However, we cannot reject the hypothesis that there are different and distinct antigenic variants of Eurasian H14 viruses that have yet to be found in nature. Additionally, we found that all of the studied viruses—H14N3, H14N9, and H4N6—were able to replicate efficiently in MDCK cells with or without the presence of trypsin, and were highly sensitive to oseltamivir (Tamiflu) in experiments. These data allow us to expect that, in the future, we will be able to search and cultivate more isolates of this rare subtype.

Interspecies transmission usually occurs as a result of adaptation of influenza strains to the receptor specificity of the tissues of the new host. In a series of experiments in the 1980s, Paulson and colleagues found that avian and equine influenza viruses bind predominantly to sialic acids, characterized by an α2-3-glycosidic bond with galactose, while human influenza viruses bind to α2-6-SA receptors [29].

The study of the receptor-binding phenotypes of avian influenza viruses of various subtypes makes it possible to trace evolutionary transitions between different hosts, while also making it possible to establish the necessary minimum of receptor adaptations to overcome interspecies barriers. Our study showed that viruses of the H14 subtype effectively bind to receptors containing fucose. Gambaryan et al. established that influenza viruses of different subtypes that circulate in poultry usually had high binding avidity to receptors containing fucose or sulfate in the last part of the molecule—GlcNAc-3 [23]. However, duck influenza viruses, with the H4 subtype being that which circulates most widely in these species, do not bind to receptors that contain fucosyl moieties. Since, in our case, sulfate-bound fucose was located in the middle part of the molecule, this may be a new receptor pattern for H14-subtype influenza viruses circulating in ducks.

The isolated viruses were also tested with different variants of the structures of sialic acids containing Neu5Gc. According to the published data, equine and duck influenza viruses bind to the glycolyl form of neuraminic acid [30,31]. However, in our study, duck viruses of the H14 and H16 subtypes did not bind to Neu5Gc-based receptors.

Data suggest that, although avian influenza viruses preferentially bind to receptors that possess a 2-3 bond between acetylneuraminic acid and galactose, there are differences in receptors depending on bird species.

The presence of the H14 subtype in Eurasia for the last 7 years has been exclusively associated with the territories of Western Siberia. However, despite the dramatic relatedness of internal segments of this subtype (PB1, PB2, MP, NP) with isolates from Bangladesh, Mongolia, and China (Central Asian Flyway/CAF) and active AIV surveillance on these territories, no H14-subtype virus isolations have been reported there thus far. Therefore, we hypothesize that the reservoir host is not connected with CAF. An alternative option might be the circulation of H14 viruses, presumably in bird hosts following the Black Sea/Mediterranean flyway. Future analysis of these territories could provide a list of species for surveillance. Interestingly, the vast majority of known H14 viruses were isolated from dabbling ducks of a single genera—*Anas* sp.—in both hemispheres, mainly from teals. Our three viruses (A/29, A/210, A/211) were acquired from common teal and garganeys during the fall season. Of the currently described H14 viruses isolated in Siberia, there is only one isolation in which the H14 virus was obtained from sandpiper [7]. At the same time, most of the ring recoveries in the described region of common teals and garganeys have been associated with migrations in the westward direction and the Black Sea/Mediterranean and Central Asian flyways [32].

Previous reports on H14-subtype viruses circulating in North America suggest the introduction of AIVs from the Eurasian lineage into the New World followed by reassortments with North American variants [10]. However, phylogenetic analysis showed that no evidence has been found to date detecting the reintroduction of H14 avian influenza viruses from America to the Eurasian continent, and the recent Eurasian clade of H14-subtype viruses diverged from the most recent common ancestor with the clade introduced in North America at the beginning of the 2000s.

We found that particular internal segments of this subtype are closely related to internal segments of the recent HPAI variants, and that viruses can exchange bysegments, and can potentially play a role in HPAI circulation. Moreover, the fundamental aspects of underlying evolutionary events and hidden circulation of H14Nx viruses require further investigation. As an example, the Tomsk strain PB2 segment (Figure 5C) (yellow circle) shares a common ancestor with the PB2 segments of the H5N1 strains isolated later in Europe. Of course, internal segments are mainly considered not to be associated with pathogenicity. At the same time, a lot of HPAI (external)-LPAI (internal) reassortment variants are likely generated during outbreaks of HPAI variants, assuming some benefits for virus replication, dissemination, and/or immune evasion.

In the present work, we identified a subclade of the HA segment for Eurasian H14 viruses during the period 2014–2019 and classified them into two groups. The limitation for such subclade assignment is obviously the extremely small number of currently known sequences. However, in further studies of H14-subtype viruses with increasing numbers of available sequences, the genetic topology will be refined and detailed.

## Figures and Tables

**Figure 1 viruses-15-00734-f001:**
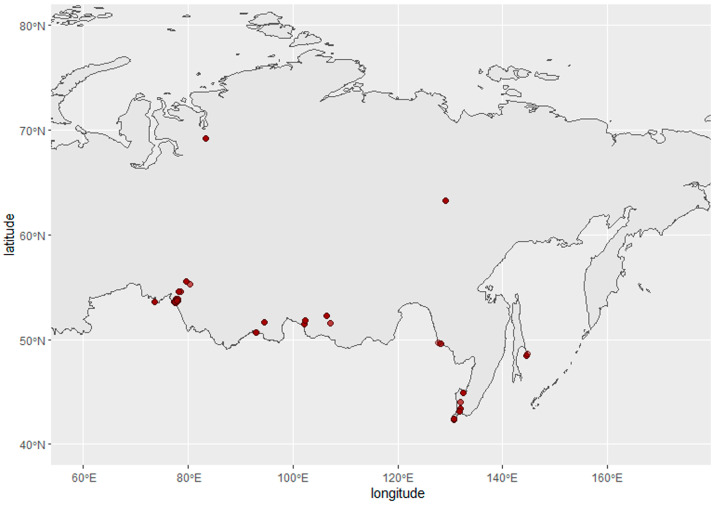
Avian influenza virus surveillance sampling sites in Russia, 2014–2019. Red circles represent sampling sites.

**Figure 2 viruses-15-00734-f002:**
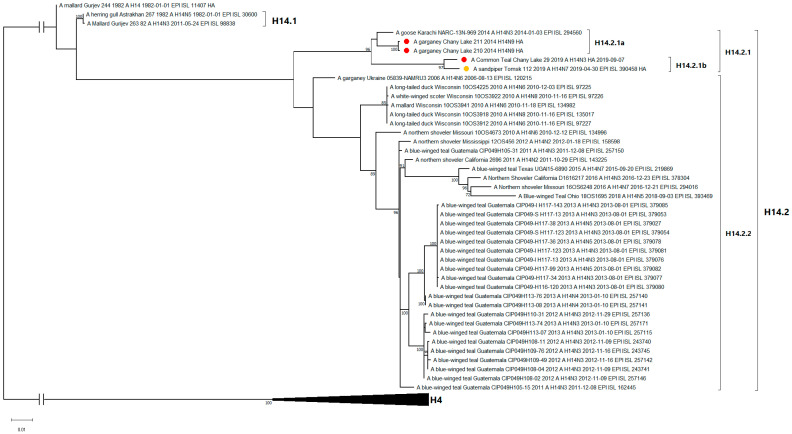
Maximum likelihood phylogenetic tree of HA subtype H14 genome segment of avian influenza viruses (●—sequences from this study; ●—sequence of H14 influenza virus strain isolated in related region from public database).

**Figure 3 viruses-15-00734-f003:**
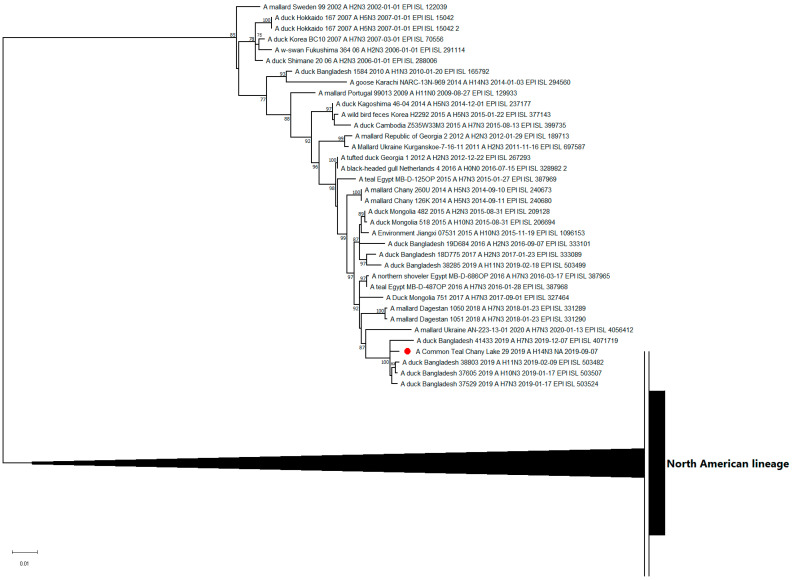
Maximum likelihood phylogenetic tree of the NA-subtype N3 genome segment of avian influenza viruses (●—sequence from this study).

**Figure 4 viruses-15-00734-f004:**
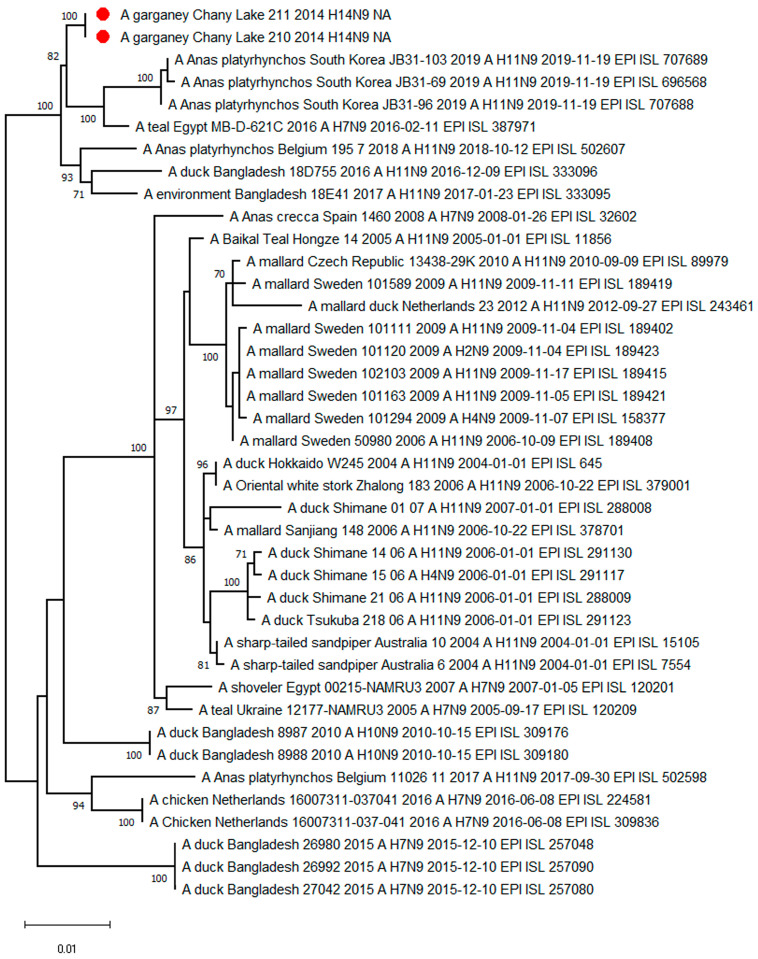
Maximum likelihood phylogenetic tree of the NA-subtype N9 genome segment of avian influenza viruses (●—sequences from this study).

**Figure 5 viruses-15-00734-f005:**
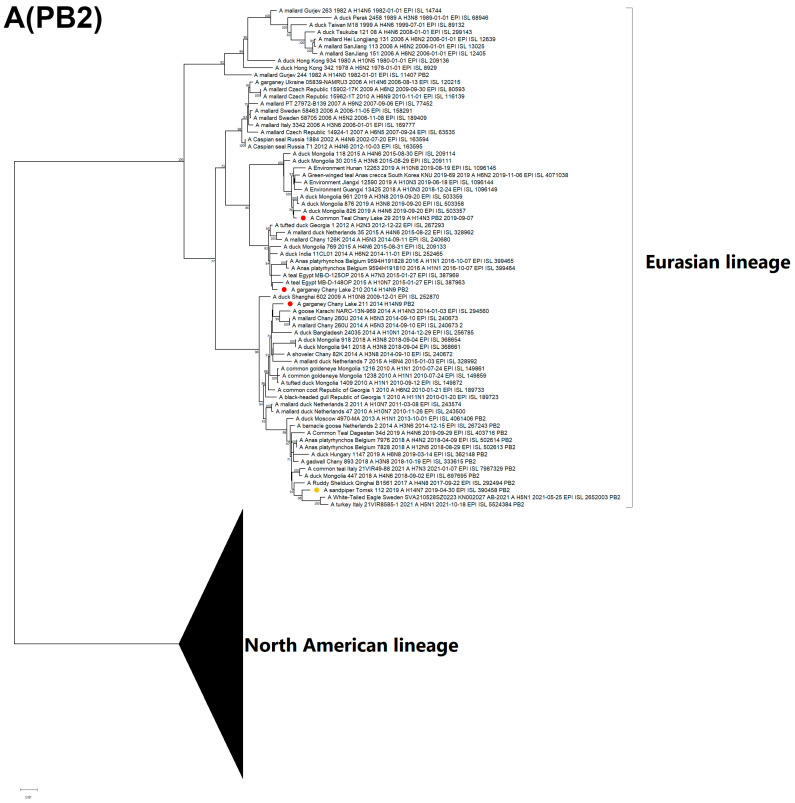
Maximum likelihood phylogenetic tree of internal genome segments: (**A**) PB2, (**B**) PB1, (**C**) PA, (**D**) NP, (**E**) MP of avian influenza viruses (●—sequences from this study; ●—sequence of H14 influenza virus isolated in related region from public database).

**Table 1 viruses-15-00734-t001:** H14-subtype AIV strain information.

Strain	Lab ID	Accession ID
A/common_teal/Chany_Lake/29/2019 (H14N3)	A/29	400267
A/garganey/Chany_Lake/210/2014 (H14N9)	A/210	14854178
A/garganey/Chany_Lake/211/2014 (H14N9)	A/211	14853905

**Table 2 viruses-15-00734-t002:** Virus titers in MDCK cells and neuraminidase enzyme inhibition assay.

Viruses	MEM with 2% FBS, log_10_TCID_50_/mL	MEM with 0.2% BSA and 2 µg/mL of Trypsin, log_10_TCID_50_/mL	Oseltamivir (Tamiflu)	
			Mean IC_50_ ^a^ (nM) ± SD	Fold Change ^b^
A/29 (H14N3)	6.55 ± 0.3	7.67 ± 0.3	0.58 ± 0.03	1.42
A/210 (H14N9)	7.17 ± 0.3	8.17 ± 0.2	0.67 ± 0.15	1.63
A/15O (H4N6)	6.30 ± 0.2	7.30 ± 0.4	0.51 ± 0.04	1.24
A(H1N1)pdm09	n/a	7.25 ± 0.3	0.41 ± 0.04	1.0

Note: ^a^ IC50, half-maximal inhibitory concentration. The IC50 denotes the concentration of a NA inhibitor that reduces the NA activity by 50% relative to NA activity without the inhibitor. NA inhibition assay used MUNANA as substrate at a final concentration of 100 µM. Values are the mean of two or three independent determinations. ^b^ Fold change relative to the mean IC50 of the A (H1N1)pdm09 virus. Fold-change values of each NA were interpreted using criteria established by the World Health Organization Influenza Antiviral Working Group. ^b^ Fold change > 10 (including reduced/highly reduced inhibition).

**Table 3 viruses-15-00734-t003:** Antigenic analysis of influenza A/H14 viruses (HI (VN) titers).

Quail Post Infectious Sera	Antigens		
	A/29 (H14N3)	A/210 (H14N9)	A/15O (H4N6)
A/29 (H14N3)	160 (80)	160 (80)	5 (5)
A/210 (H14N9)	160 (160)	160 (160)	5 (5)
A/15O (H4N6)	5 (5)	5 (5)	160 (320)

**Table 4 viruses-15-00734-t004:** Receptor binding glycoprofile of avian influenza viruses isolated from ducks and gulls.

Virus Strain	Oligosaccharides	RFU
A/garganey/Chany lake/210/2014 (H14N9)	Neu5Aca2-3Galb1-4(2-O-Su-Fuca1-3)(6-O-Su)GlcNAcb-sp3 (Neu5Aca3′(2-suFuca3)(6-su)LN-C3)	17,793
	Neu5Aca2-3(6-O-Su)Galb1-4(6-O-Su)GlcNAcb-sp3 (Neu5Aca3′(6,6′-su2)LN-C3)	17,396
A/common teal/Chany Lake/29/2019 (H14N3)	Neu5Acα2-3Galβ1-4(Fucβ1-3)GlcNAcβ-sp3 (SiaLeX)	64,732
	Neu5Acα2-3Galβ1-4(2-O-Su-Fucα1-3)GlcNAcβ-sp3	61,445
	(SiaLeX2‴Su)	
	Neu5Acα2-3Galβ1-4-(6-O-Su)GlcNAcβ-sp3	57,287
	(3′SLN6Su)	
	Neu5Acα2-3Galβ1-3(Fucα1-4)GlcNAcβ-sp3(SiaLea)	56,604
	Neu5Acα2-3Galβ1-3GlcNAcβ-sp3 (3′SiaLeC)	51,191
	Neu5Acα2-3Galβ1-4GlcNAcβ-sp3(3′SLN)	48,444
	Neu5Acα2-3GalNAcα-sp3 (3-SiaTn)	29,289
A/little tern/Guriev/779/83 (H16N3)	Neu5Acα2-6(Galβ1-3)GalNAcα-sp3 (6SiaTF)	43,606
	Neu5Acα2-3Galβ1-4(Fucβ1-3)GlcNAcβ-sp3 (SiaLeX)	29,451
	Neu5Acα2-6Galβ1-4GlcNAcβ-sp3 (6′SLN)	28,472
	Neu5Acα2-6Galβ1-4-(6-O-Su)GlcNAcβ-sp3 (6′SLN6Su)	27,473
	Neu5Acα2-3Galβ1-4GlcNAcβ-sp3 (3′SLN)	15,806

**Table 5 viruses-15-00734-t005:** Nucleotide identity of the HA segment of Siberian H14Nx viruses.

Strain	Related Strain	Country	HA Identity, %
A/29	A/sandpiper/Tomsk/112/2019 (H14N7)	Russia	98.65
A/210	A/goose/Karachi/NARC-13N-969/2014 (H14N3)	Pakistan	98.24
A/211	A/goose/Karachi/NARC-13N-969/2014 (H14N3)	Pakistan	98.24

**Table 6 viruses-15-00734-t006:** Amino acid substitutions of hemagglutinin protein of H14-subtype AIVs.

Strain	6	58	62	77	127	143	145	189	211	226	228	281	301	326	329	441	515	523	Cleavage Site Amino Acid Sequence
A/goose/Karachi/NARC-13N-969/2014 (H14N3)	I	I	N	H	G	R	G	Q	I	Q	G	P	I	D	A	T	I	M	NIPGKQAK/G
A/common teal/Chany Lake/29/2019 (H14N3)	T	.	D	.	N	.	S	T	T	Q	G	S	.	.	T	.	V	.	NIPDKQTK/G
A/garganey/Chany Lake/210/2014 (H14N9)	T	V	D	.	S	.	.	.	.	Q	G	.	.	.	.	.	V	.	NIPDKQAK/G
A/garganey/Chany Lake/211/2014 (H14N9)	T	V	D	.	S	.	.	.	.	Q	G	.	.	G	.	.	V	.	NIPGKQAK/G
A/sandpiper/Tomsk/112/2019	T	.	D	R	N	H	S	K	T	Q	G	.	M	.	T	A	V	T	NIPDKQTK/G

## Data Availability

All sequences from the study are available in GISAID EpiFlu database (accession numbers: 400267, 14854178, 14853905).

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
