# Peer review of "Virological and Genetic Characterization of the Unusual Avian Influenza H14Nx Viruses in the Northern Asia"

_viruses, 2023, doi:10.3390/v15030734_

Round 1

Reviewer 1 Report

The paper, " Genetic characterization of the rare identified avian influenza H14Nx viruses in the Northern Asia" by Nikita et al. describes a genetic characterization of H14Nx viruses in Northern Asia. In this paper, a new subtype of avian influenza virus (H14N9) was isolated, and the low isolation rate of the H14Nx virus is described. Although the genetic evolution of H14Nx is described in detail in this paper, other biological characteristics of H14Nx viruses are not well known at present. Therefore, please give a further revision as followings:

Major issues:

To better understand the characteristics of H14Nx viruses, some in vivo and in vitro tests could be added to better understand of H14Nx viruses.

Minor issues:

1.      Line 110-111: Please provide documentation of the separation rates for each subtype.

2.      Line 226 and Table 3: Please analyze amino acid substitutions of hemagglutinin protein that affect the phenotype, such as increasing virus binding to α2-6 and so on (PMID: 17108965, PMID: 20041223, PMID: 34686117, PMID: 28814518….). Please cite the direct reference. In addition, the table does not show amino acid signature residues 226 and 228 mentioned in the text.

3.      Line 29: Please confirm whether N10 and N11 have been isolated from wild waterfowl and cite relevant references correctly.

4.      Line 126-127 and Table 1: These strains need to be named consistently.

5.      Figure 5: The internal gene ML trees should be sorted according to their molecular size. And why doesn't the ML tree of MP genes use triangles like other ML trees?

6.      Line 29: “was detected” should be “were detected”

7.      Line 280: “internal segments considered to be” should be “internal segments are considered to be”

Author Response

Response to Reviewer 1 Comments

Dear Reviewer,

Thank you for your supportive feedback and useful suggestions to improve the manuscript. We have revised the manuscript by additional experiments providing and modifying Text, Figures, and Tables based on the comments.

Major issues:

To better understand the characteristics of H14Nx viruses, some in vivo and in vitro tests could be added to better understand of H14Nx viruses.

Response:

We conducted additional experiments to provide a biological characterization of the H14 isolates. In particular, we conducted hemagglutination inhibition and virus neutralization assays, estimated the susceptibility of isolates to neuraminidase inhibitors, and characterized receptor specificity. We found that A/29 and A/210 virus isolates being in different phylogenetic subclades are antigenically similar, but did not have significant cross-reaction with the H4-reference strain confirming the difference from the sister subtype. We confirmed the sensitivity of the isolates to oseltamivir and the efficient replication of viruses in MDCK cells with and without trypsin. We corrected the aim of the research; added the description of methods, results and discussion.

Minor issues:

  1. Line 110-111 (Revised manuscript line 189-190): Please provide documentation of the separation rates for each subtype.

Response 1:

We are thankful for the suggestion to improve the manuscript. The primary purpose of the research was to detect, isolate and analyze the genetic characteristics of H14 subtype viruses, which have a very low presence in wild bird populations. To provide evidence of the low detection rate we described the most common influenza virus subtypes prevalence in this region. In the future, we aim to analyze long-term surveillance data to provide a comprehensive study on subtype prevalence, dynamics, and genetic diversity of avian influenza viruses in the Asian part of Russia.

  1. Line 226 and Table 3 (Revised manuscript line 374 and Table 6): Please analyze amino acid substitutions of hemagglutinin protein that affect the phenotype, such as increasing virus binding to α2-6 and so on (PMID: 17108965, PMID: 20041223, PMID: 34686117, PMID: 28814518….). Please cite the direct reference. In addition, the table does not show amino acid signature residues 226 and 228 mentioned in the text.

Response 2:

Thank you for the suggestion. We edited Table 6 according to the comment. Hemagglutinin protein amino acid substitutions known to affect the phenotype are mostly investigated for HPAI viruses of the H5 subtype. Amino acid identity between H5 and H14 hemagglutinin protein is low and these subtypes are in highly diverged phylogenetic groups (H5 in group 1 and H14 in group 2) (PMID: 19004788). Following the comment, we analyzed amino acid substitutions known to affect phenotype. For example, N/S123, S182, and K192 amino acids (H5 numbering; N/S127, S186, K196 for H14 numbering) in our isolates are not the same as in reference H5N1 strain (A/mallard/Italy/3401/2005 H5N1: N123, N182, Q192). Amino acid substitutions of these sites are known to affect the phenotype of the H5N1 virus (PMID: 17108965). Based on the phylogeny and the low level of identity of HA proteins of the H5 and H14 subtypes, we believe that it is hard to infer the effect of particular substitutions known for the H5 subtype; studies involving reverse genetics for the H14 subtype viruses are needed.

  1. Line 29 (Revised manuscript line 33): Please confirm whether N10 and N11 have been isolated from wild waterfowl and cite relevant references correctly.

Response 3:

Thank you for this notification. N10 and N11 subtypes are known to be detected in bat populations only. We edited this sentence and provided relevant references for subtypes detection.

  1. Line 126-127 and Table 1 (Revised manuscript lines 212-213): These strains need to be named consistently.

Response 4:

We are thankful for this comment. We have edited Table 1 following the nomenclature system (WHO Memorandum, 1980) (PMID: 6969132)

  1. Figure 5 (Revised manuscript Figure 5): The internal gene ML trees should be sorted according to their molecular size. And why doesn't the ML tree of MP genes use triangles like other ML trees?

Response 5:

We appreciate the reviewer for their comment. We have revised the figure and sorted it and the description according to the molecular size of AIVs genome segments. The width of the triangle in the North American clade depicts the distance (based on the number of substitutions per site) from the node of the clade to the most diverged leaf. In our data set for the North American clade of the MP segment this distance is low.

  1. Line 29 (Revised manuscript line 33): “was detected” should be “were detected”

Response 6:

Thank you for this comment. We edited the text according to the comment.

  1. Line 280 (Revised manuscript line 477): “internal segments considered to be” should be “internal segments are considered to be”

Response 7:

We corrected “internal segments are considered to be” instead of “internal segments considered to be” according to the comment.

Reviewer 2 Report

Dear authors

Minor editing is suggested, as follows:

L 2-3 The title can be improved. An option could be "Genetic characterization of the unusual H14 avian influenza subtypes isolated from wild birds in Siberia, 2014-2019"

L6-8 The emails of the co-authors are missing.

L39-40 Please use brackets instead of parenthesis after "gull" and before "[4].

L43 Please delete space before "in 2019 [5].

L66 Please lowercase for "Specific Pathogen Free"

L70. 78, 87 Please include the city of the manufacturer also.

L71-72 Please delete "Russia" after "Amplisens"

L84-85 Please clarify and specify how A/29 strain library preparation was performed and the Department from the NIAH from Tsukuba, Japan, which was involved in this work. 

L89 Please delete "USA" and change from "with the help" to "by using".

L04 Please add a comma to change from "5777" to " 5,777".

L113 Please add a comma after "204".

L118 Please delete space before "Contribution".

L145, 161 Please add "the former" before "USSR".

L151 Please delete "(Fries et al.," and add a parenthesis to close "2103".

L156 Please add "(2013)" after "et al."

L166 Please add "In this study," after "(Appendix, S3)." and use past tense for "belong"

L175-178 Please rewrite this paragraph

L189 Please add "respectively" after "Bangladesh"

L191 Please add "from the period between" before 2015

L199 Please add "in" between "Pakistan" and "2014"

L241 Please delete "(Fries et al., 2013)" and explain that this hypothesis came from a previously published paper.

L252 Please complete this sentence.

L277 Please clarify the term "cagey". Is it related to the meaning of this paragraph?

Author Response

Response to Reviewer 2 Comments

Dear Reviewer,

Thank you for your useful comments and constructive suggestions which helped us to improve the manuscript. Additional experiments were provided for biological characterization of the isolates according to Reviewer 1 comments. We edited text according to comments and suggestions. Please see below detailed response on each point of the review and corrections of the manuscript.

Minor issues:

  1. L 2-3 (Revised manuscript L 2-3). The title can be improved. An option could be "Genetic characterization of the unusual H14 avian influenza subtypes isolated from wild birds in Siberia, 2014-2019"

Response 1:

We are thankful for the suggestion to improve the manuscript title. We replaced “rare-identified” with “unusual”

  1. L6-8 (Revised manuscript L 6-8) The emails of the co-authors are missing.

Response 2:

Thank you for the comment. Author list and affiliations provided in the text made according to Instructions for Authors and manuscript template, only corresponding author email provided in the manuscript file. Emails of the co-authors added via web-based submission form.

  1. L39-40 (Revised manuscript L 42-45). Please use brackets instead of parenthesis after "gull" and before "[4]

Response 3:

Thank you for comments and suggestions. We used brackets instead of parenthesis.

  1. L43 (Revised manuscript L 48). Please delete space before "in 2019 [5].

Response 4:

We removed space according to the comment.

  1. L66 (Revised manuscript L 72). Please lowercase for "Specific Pathogen Free"

Response 5:

We corrected the text following the comment.

  1. L70. 78, 87 (Revised manuscript L 76, 156, 165). Please include the city of the manufacturer also.

Response 6:

We have included the city of the manufacturer.

  1. L71-72 (Revised manuscript L 76, 77). Please delete "Russia" after "Amplisens"

Response 7:

Thank you for this comment. We removed country of the manufacturer according to the comment.

  1. L84-85 (Revised manuscript L 162-163). Please clarify and specify how A/29 strain library preparation was performed and the Department from the NIAH from Tsukuba, Japan, which was involved in this work. 

Response 8:

We added the reference to the published work with the same protocol used and provided more details. We provided the information of the department of the NIAH involved in this work.

  1. L89 (Revised manuscript L 168). Please delete "USA" and change from "with the help" to "by using".

Response 9:

We deleted country of the manufacturer from the text and replaced "with the help" with "by using".

  1. L04 (Revised manuscript L 184). Please add a comma to change from "5777" to " 5,777".

Response 10:

We edited the number according to the comment.

  1. L113 (Revised manuscript L 193). Please add a comma after "2014".

Response 11:

We added comma following the comment.

  1. L118 (Revised manuscript L 198). Please delete space before "Contribution".

Response 12:

We removed space before “Contribution”.

  1. L145, 161 (Revised manuscript L 278, 294). Please add "the former" before "USSR".

Response 13:

We add “the former” before “USSR”.

  1. L151 (Revised manuscript L 284). Please delete "(Fries et al.," and add a parenthesis to close "2103".

Response 14:

We edited line according to the comment.

  1. L156 (Revised manuscript L 290).  Please add "(2013)" after "et al.".

Response 15:

We added the year in round brackets after "et al.".

  1. L166 (Revised manuscript L 299).  Please add "In this study," after "(Appendix, S3)." and use past tense for "belong".

Response 16:

We corrected the line following the comment

  1. L175-178 (Revised manuscript L 308-311). Please rewrite this paragraph.

Response 17:

We rewrite the paragraph to clarify phylogenetic relations of nucleotide sequences of NA segment of N3 subtype

  1. L189 (Revised manuscript L 356). Please add "respectively" after "Bangladesh".

Response 18:

Thank you for your suggestion. The list of countries here does not strictly match the order of the list of subtypes.

  1. L191 (Revised manuscript L 359). Please add "from the period between" before 2015.

Response 19:

We added "from the period between" before 2015

  1. L199 (Revised manuscript L 331). Please add "in" between "Pakistan" and "2014".

Response 20:

We added “in" between "Pakistan" and "2014".

  1. L241 (Revised manuscript L 405). Please delete "(Fries et al., 2013)" and explain that this hypothesis came from a previously published paper.

Response 21:

We removed reference in parentheses and edited the text according to the comment with reference addition in brackets.

  1. L252 (Revised manuscript L 410). Please complete this sentence.

Response 22:

We appreciate the reviewer for the comment. We completed the sentences with the addition of “subtype avian influenza viruses”.

  1. L277 Please clarify the term "cagey" (Revised manuscript L 474). Is it related to the meaning of this paragraph?

Response 23:

Thank you for this comment. To clarify the meaning of the sentence we replaced “”cagey” evolution” with “underlying evolutionary events and hidden circulation”. H14 subtype viruses have low detection rate, while it circulates, evolve and reassort unseen most of the time. Following the phylogenetic reconstruction we can see internal segments, closely related to HPAI variants.

Round 2

Reviewer 1 Report

The revision is good for publication.